# Immunotherapy with Monoclonal Antibodies for Acute Myeloid Leukemia: A Work in Progress

**DOI:** 10.3390/cancers15205060

**Published:** 2023-10-19

**Authors:** Matteo Molica, Salvatore Perrone, Costanza Andriola, Marco Rossi

**Affiliations:** 1Department of Hematology-Oncology, Azienda Universitaria Ospedaliera Renato Dulbecco, 88100 Catanzaro, Italy; rossim@unicz.it; 2Department of Hematology, Polo Universitario Pontino, S.M. Goretti Hospital, 04100 Latina, Italy; sperrone@hotmail.it; 3Hematology, Department of Translational and Precision Medicine, Sapienza University, 00100 Rome, Italy; costanza.andriola@uniroma1.at

**Keywords:** acute myeloid leukemia, immunotherapy, checkpoint inhibitors, new therapeutic landscape, bispecific antibodies

## Abstract

**Simple Summary:**

Immune based treatments (ITs) represent one of the most important strategies in the treatment of acute myeloid leukemia (AML), like allogeneic stem cell transplant. Recently, several strategies have been explored: monoclonal antibodies (immunoconjugates or not, checkpoint inhibitors), bispecific antibodies (BiTE), vaccination, and chimeric antigen-receptor (CAR) T cells. This review will mainly focus on check-point inhibitors and BiTE, despite none of these being currently approved for patients with AML. The reasons for the struggle in the application of these drugs will be analyzed.

**Abstract:**

In the last few years, molecularly targeted agents and immune-based treatments (ITs) have significantly changed the landscape of anti-cancer therapy. Indeed, ITs have been proven to be very effective when used against metastatic solid tumors, for which outcomes are extremely poor when using standard approaches. Such a scenario has only been partially reproduced in hematologic malignancies. In the context of acute myeloid leukemia (AML), as innovative drugs are eagerly awaited in the relapsed/refractory setting, different ITs have been explored, but the results are still unsatisfactory. In this work, we will discuss the most important clinical studies to date that adopt ITs in AML, providing the basis to understand how this approach, although still in its infancy, may represent a promising therapeutic tool for the future treatment of AML patients.

## 1. Introduction

Anti-cancer immune-based therapeutics (ITs) rely on different approaches that aim to support host immune effectors in clearing neoplastic clones. Although ITs have improved the prognosis of several advanced-stage solid tumors (metastatic melanoma, renal cell carcinoma, head and neck cancers, and non-small lung cancer) [1], the benefits of ITs have been limited in relation to hematological malignancies, and are mostly restricted to classical Hodgkin lymphoma and primary mediastinal B cell lymphoma [2,3,4]. In the context of acute myeloid leukemia (AML), despite the introduction of new drugs and the presence of three updated classifications published in 2022 [5,6], which allow for more effective prognostic stratification, no ITs drugs are currently approved. Several considerations need to be made in order to correctly evaluate the role and pitfalls of Its; for example, AML patients are severely immuno-compromised, and the introduction or addition of novel therapies may potentially alter the landscape of infectious complications, with guidelines for clinical practice being frequently updated by the European Conference on Infections in Leukemia (ECIL) [7,8]. Similarly, AML- and therapy-related damage can hamper the immunological clearance of AML blasts by natural killer (NK) and CD8+ T cells [9]. Furthermore, AML blast cells can evade the immune system by reducing either the expression and/or the presentation of HLA-restricted leukemia-associated/specific antigens (e.g., MHC class II genes loss or downregulation after allogeneic-SCT has been extensively studied [10]). Moreover, bone marrow-infiltrating T lymphocytes, although numerically similar to healthy donors, cause the upregulation of checkpoint inhibitors such as PD1, TIM-3 and LAG-3 [11]. In a murine model of AML, CD8+ T cells were shown to present a typical exhaustion phenotype (TIM-3/PD-1-positive), whereby the expression of TIM-3/PD-1 increases during AML progression and can be partially reduced by anti-PD1 treatment [12]. In addition, AML blasts promote a tolerogenic immune microenvironment by releasing reactive oxygen species (ROS), arginase (Arg), indoleamine 2,3-dioxygenase (IDO), and extracellular vesicles into the niche. These mediators can specifically inhibit the cytotoxic activity of T and NK cells. Moreover, they lead to the exhaustion of T cells, regulatory T cells (Tregs) and myeloid-derived suppressor cells (MDSCs), while promoting the switch of tumor-associated macrophages (TAMs) from the immunogenic M1 to the immune-suppressive M2 phenotype [13]. Indeed, TAMs can exert a dual function as part of the tumor-extrinsic pathways of both primary and adaptive resistance to Its, by expressing several immunosuppressive molecules such as checkpoint ligands (e.g., PDL-1, PDL-2, CD80, and CD86) [14]. Moreover, TAMs can promote the apoptosis of neoplastic cells activated by the stimulator of interferon genes (STING), CD40, as well as the engagement of toll-like receptors [14]. Even considering these limitations, ITs represent an area of intensive clinical research.

ITs in AML comprise at least four categories of experimental treatments (Figure 1): (A) allogeneic-SCT, consisting of polyclonal lymphocytes derived from the donor that exert graft vs. leukemia effects and provide stem cells that can repopulate the host’s bone marrow; (B) monoclonal antibodies (MoAb), such as chemo-immunoconjugates (e.g., gemtuzumab ozogamicin [15]), several chec-point inhibitors, and bispecific antibodies (BiTE); (C) IL-2 maintenance [16] and vaccination [17]; and (D) chimeric antigen–receptor (CAR) T cells [18,19].

In the present work, we will outline the state of art of studies employing MoAb to fight AML, highlighting the strategies adopted to overcome its natural resistance to ITs (Figure 2).

## 2. Immune Checkpoint Inhibition in Acute Myeloid Leukemia

### 2.1. PD-1/PDL-1 Blockade

The PD-L1/PD-1 axis has been thoroughly studied over the past few decades, and recent data indicate that CD34+ blasts from MDS and AML patients cause the upregulation of PDL-1, while PD-1 expression is raised in effector T cells and Tregs [11,20,21,22]. Furthermore, activated T cells upregulate the antagonistic co-receptor PD-1. As the PDL-1/PD-1 interaction dampens anti-tumor T cell responses [23,24,25], the disruption of this pathway by anti-PD-1/PDL-1 monoclonal antibodies can revive worn-out T cells and prompt anti-tumor responses [26,27]. Although this strategy has provided striking data in the context of solid malignancies, its action in the context of AML and MDS needs to be further clarified (Table 1).

### 2.2. PD-1 Inhibitors

#### 2.2.1. Nivolumab

In a phase II study, 70 patients with R/R AML were treated with a combination of nivolumab (3 mg/Kg) on days 1 and 14 as well as azacitidine at 75 mg/m^2^ for one week, every 4 to 6 weeks. The median age of patients was 70 years (range: 22–90); the median number of previous therapies was two (range: one to seven). The overall response rate (ORR) was 33%, with 7 patients reaching hematologic improvement that lasted for more than 6 months and 15 (22%) achieving full remission or complete remission with insufficient count recovery. Six patients (9%) showed stable disease for more than six months. When comparing hypomethylating agent (HMA)-naive patients (*n* = 25) and HMA-pretreated patients (*n* = 45), the derived ORR values were 58% and 22%, respectively. Eight (11%) patients experienced immune-related side effects graded 3 to 4. The pretreatment of bone marrow and peripheral blood with CD3+ and CD8+ T cells strongly predicted the response. Interestingly, after two and four doses of the drug, non-responder patients showed increased CTLA-4 expression on CD4 T cell effectors [28]. On this basis, the protocol was amended, with the introduction of a second cohort, wherein 31 R/R AML patients with high CTLA-4 expression were treated with anti-CTLA-4 moAb ipilimumab, together with azacitidine and nivolumab, to further improve T cell responses. With a median OS of 10.5 months, the ORR rate among patients for whom efficacy could be evaluated (*n* = 24) was 46% (CR/CRi rate: 36%), which compares favorably with the result for azacitidine plus nivolumab. As anticipated, 25% of patients experienced grade 3–4 immune-related adverse events (ir-AEs), which is higher than that seen in the azacitidine plus nivolumab cohort [29].

Nivolumab was also investigated by Davids et al., who focused on recurrent hematologic malignancies following allogeneic transplantation (10 AML cases out of 28). This multicenter phase I trial evaluated the effectiveness, the immunologic activity, the maximum tolerated dose and the safety of the drug. Nivolumab was given out every two weeks until the occurrence of either progression or intolerable side effects. The initial dose level was 1 mg/kg, with the option to reduce it to 0.5 mg/kg or increase it to 3 mg/kg. The 1-year PFS rate was 23% and the OS rate was 56%, with a median follow-up of 11 months. Among the 28 patients enrolled, 3 patients could not be evaluated for the efficacy of response, whereas 25 patients could be, owing to early toxicity. Of the 25, 20 received treatment at 0.5 mg/kg, while 5 received treatment at 1 mg/kg. As regards the efficacy at both levels, the ORR was 32%. The median OS for all patients was 21.4 months, and the overall patient 1-year OS rate [30] was 56%. Two of the six patients treated at 1 mg/kg experienced dose-limiting toxicity (DLT) from ir-AEs. In total, 22 patients were treated at 0.5 mg/kg, and four DLTs occurred, including two ir-AEs and two fatal GVHD.

A phase 2 trial tested the use of nivolumab in combination with idarubicin + cytarabine in frontline disease treatment in patients with newly diagnosed AML (*n* = 42) and HR-MDS (*n* = 2). The composite CR rate was 78%, of which 79% showed no evidence of measurable residual disease (MRD), assessed using multiparameter flow cytometry (MFC). In total, 19 patients underwent allo-SCT, with 13 patients developing GVHD (68%). The chemo-immunotherapy regimen showed a tolerable toxicity without the extensive incidence of ir-AEs. Overall, the median OS was 18.5 months, and for those who received allo-SCT, this was prolonged to 25 months. Importantly, responders who were continued on nivolumab treatment after remission and those who were bridged to allo-SCT showed similar OS values, suggesting that nivolumab may play a role in restoring anti-tumor immune surveillance and eliminating MRD [31].

As confirmed by observations in certain lymphoma treatment subgroups, Liu et al. [31] postulated that immunomodulation with checkpoint inhibitors (CIs) might induce an anti-leukemia immune response avoiding or postponing a potential disease relapse in intermediate/poor-risk AML. NCT02275533 comprised a phase II trial to assess maintenance using nivolumab in patients with AML in first CR or CR with incomplete hematologic recovery (CRi), as assessed by bone marrow biopsies within 2 months from the last chemotherapy, and who were not eligible for HSCT. Eighty patients were randomized to placebo or nivolumab treatment (3 mg/kg every 2 weeks for forty-six doses). The 2-year PFS values were identical—30.3% and 30% in the nivolumab arm and the observation arm, respectively (*p* = 0.38). The median OS values were 53.9 and 30.9 months in the two subgroups; however, the 2-year OS values were 60.0% and 52.8% in the nivolumab and observation arm, respectively (*p* = 0.23). As expected, AEs were more common in the nivolumab arm, with 27 (71.0%) experiencing grade 3 or higher toxicity, while only 5 patients in the placebo group showed the same (12.2%) (*p* < 0.001) [32]. These findings show that nivolumab maintenance did not benefit high- and intermediate-risk AML patients after chemotherapy, as it failed to improve the 2-year PFS or the OS, and it increased the incidence of AEs.

To describe transplantation outcomes with regard to GvHD and the effects of various GvHD prophylaxis strategies, Oran and colleagues conducted a clinical trial of patients with AML and MDS patients treated with either PD-1 (nivolumab) and/or CTLA-4 (ipilimumab) inhibitors before undergoing an allo-HSCT. Of the 43 patients enrolled, 9 received ipilimumab and 34 received nivolumab. Nivolumab was administered to the patients in 2 cases as a monotherapy, in 19 cases in combination with chemotherapy, and in the remaining cases in combination with HMAs. Of the 43 patients, 24 experienced CR, 6 experienced CRi, 5 showed hematologic improvement, and 1 experienced a PR. At the time of best response, 27 of these patients discontinued CI therapy and proceeded to allo-HSCT. Tacrolimus and mini-methotrexate with or without anti-thymocyte globulin were used as part of the GvHD prophylaxis in 21 patients, while post-HSCT cyclophosphamide (PTCy) or tacrolimus with or without mycophenolate mofetil was used in 22 individuals. The application of prophylaxis in the patients was based on donor type. Both the baseline disease and transplantation characteristics were comparable between PTCy patients and no-PTCy patients—32% and 10%, respectively. Matched control analyses using patients with no prior exposure to Cis unfortunately showed increased aGVHD (grade 3–4) in those exposed to CI [33].

#### 2.2.2. Pembrolizumab

The combination of pembrolizumab and azacitidine was evaluated in a phase 2 trial including de novo and R/R AML. Patients received azacitidine at 75 mg/m^2^ on days 1–7, with pembrolizumab (200 mg) beginning on day 8 and every 3 weeks thereafter. Among the 37 R/R AML patients, 29 were evaluable for response and showed an ORR of 55%, with 14% CR/CRi; among the 22 patients with de novo AML who were ineligible for intensive chemotherapy, 17 were evaluable for response, and showed an ORR of 94% with 47% CR/CRi. The median OS for this cohort was 13.1 months [34]. In another trial (NCT02996474), pembrolizumab was given with 10 days of decitabine to 10 patients with R/R AML. The median OS was 10 months with responses observed in six patients: two achieved CR, one showed a morphologic leukemia-free state and three experienced stable disease [35]. A US group conducted a phase II trial providing high-dose cytarabine followed by pembrolizumab at 200 mg on day 14 to assess whether this combination could improve outcomes of R/R AML. Thirty-seven patients were enrolled, with a median age of 54 years; the majority showed either refractory (43%) or relapsed disease with a CR duration of <1 year (43%). The median OS of the entire cohort was 11.1 months, with ORR and composite CR values of 48% and 38%, respectively. Among the patients showing refractoriness/early relapse and those who received this combination approach as the first salvage therapy, there were encouraging outcomes (median OS, 13.2 and 11.3 months, respectively). Rare grade ≥ 3 ir-AEs after pembrolizumab administration included maculopapular rash (*n* = 2; 5%), aminotransferase elevation (*n* = 2; 5%), and after the failure of HMAs, lymphocytic infiltration shown on liver biopsy (*n* = 1; 3%) [36]. In a small phase 2 study conducted on 20 patients affected by intermediate-risk (60%) and high-risk AML in CR, pembrolizumab was administered as the consolidation after autologous-SCT in eight doses. The study met the planned objective of a leukemia-free survival rate of 48.4% at 2 years, and the two-year OS, NRM and CIR values were 68%, 5% and 46%, respectively [37].

The use of pembrolizumab was also investigated in high-risk and HMA-resistant MDS to define its potential efficacy in myeloid malignancies. The phase 1b KEYNOTE-013 study evaluated the use of pembrolizumab as a monotherapy in 28 patients with high-risk MDS following HMA treatment failure. No patient achieved CR or PR; 5 patients (19%) showed a complete bone marrow response, 12 (44%) showed stable disease, 10 (37%) showed progressive disease, 6 (22%) showed a cytogenetic response, and 5 (19%) showed hematologic improvement. The median OS was 6.0 months with a 2-year OS rate of 17% [38]. Another phase 2 study explored the use of pembrolizumab and azacitidine in patients with both de novo and intermediate- to high-risk MDS who did not respond to HMAs. After a median follow-up of 12.8 months, the median OS was not reached, and the ORR and CR rate were 76% and 18%, respectively, in HMA-naïve patients (*n* = 17). In the 20 patients for whom HMA treatment failed, the ORR was 25%, with a CR rate of 5% and a median OS of 5.8 months [39]. According to these data, pembrolizumab appeared to be ineffective in patients with high-risk MDS and in those who were HMA-resistant, both when used as a monotherapy and in combination regimens.

#### 2.2.3. Tislelizumab

Tislelizumab is an experimental humanized immunoglobulin G4 monoclonal antibody that has a high affinity for the PD-1 protein. It is constituted in such a way as to reduce binding to the Fc receptor on macrophages to prevent antibody-dependent phagocytosis, which is a potential mechanism of resistance to anti-PD-1 therapy [40]. Tislelizumab as a monotherapy or in combination with chemotherapy exhibited antitumor effectiveness in patients with solid tumors in early-phase clinical studies, and it also displayed a safety profile similar to thos of other anti-PD-1 antibodies [41,42].

In a phase 2, single-arm study, R/R AML patients received azacitidine or decitabine plus CAG regimen (cytarabine, aclarubicin, G-CSF) with tislelizumab. In total, 27 patients were enrolled; the ORR was 63% (14 CR/Cri, 3 PR, 10 no response). The median OS and EFS were 9.7 and 9.2 months, respectively, with grade 2–3 ir-AEs observed in four patients (14.8%) [43].

### 2.3. PD-L1 Inhibitors

The phase 2 study FUSION-AML-001 was designed to evaluate the activity and safety of durvalumab, a PDL-1 inhibitor, in combination with azacitidine in untreated patients with AML aged ≥65 years. Patients were randomized to receive azacitidine at 75 mg/m^2^ on days 1–7, either with (Arm A, *n* = 64) or without (Arm B, *n* = 65) durvalumab (1500 mg on day 1), repeated every 4 weeks. This combination regimen failed to significantly improve ORR (Arm A, 31.3%; Arm B, 35.4%), duration of response (Arm A, 24.6 weeks; Arm B, 51.7 weeks) or OS (Arm A, 13.0 months; Arm B, 14.4 months) [44]. Another PDL-1 antibody, avelumab, that has been approved by the FDA for treating Merkel cell carcinoma, renal cell carcinoma, and urothelial carcinoma was tested on untreated and R/R AML. A phase 1 study was conducted to evaluate the combination of avelumab and decitabine as the frontline treatment for AML patients unfit for intensive chemotherapy (*n* = 7). The response rate was unsatisfactory, with only one patient who reached CR and three who reached stable disease as the best responses to the therapy [45]. Avelumab was also investigated in association with azacitidine in the R/R setting. In a phase 1b/2 trial, 19 patients received azacitidine at 75 mg/m^2^ on days 1–7 in combination with avelumab on days 1 and 14, repeated every 28 days (avelumab was given at 3 mg/kg to the first 7 patients and increased to 10 mg/kg for the other 12 patients). The median age was 66 years; 100% showed adverse-risk disease (European LeukemiaNet 2017), and 63% had previously received a hypomethylating agent. The median OS was 4.8 months and responses with full or incomplete bone marrow recovery were seen in 10.5%, which result is similar to the historical CR/CRi rate of 16% observed with the application of HMA [46,47]. A further analysis in these studies confirmed that the overexpression of PD-L2 in myeloid blasts and the monocyte-restricted increase in PDL-1 expression with therapy are the most important causes of inactivity related to PDL-1 inhibitors in patients with AML [44,46].

No benefits related to the employment of anti-PDL-1 inhibitors were seen in MDS patients.

A phase 2 study addressing frontline azacitidine plus the ICI durvalumab (Arm A) versus azacitidine monotherapy (Arm B) in 84 HR-MDS patients was conducted. The ORR values were 61.9% and 47.6% in Arm A and Arm B (*p*= 0.18), respectively, with median OS values of 11.6 months and 16.7 months (*p*= 0.74), confirming the inefficacy of PDL-1 inhibitors in the MDS scenario as well [48].
cancers-15-05060-t001_Table 1Table 1Trials including PD-1 inhibitors in patients with AML.ReferenceTherapeutic ApproachType of AMLNumber of PatientsResponseSurvivalAEsDaver et al. [28]Azacitidine + nivolumabR/R AML70ORR: 33% CR/CRi: 22%Median OS: 9.2 monthsirAEs grade 3–4: 11% (*n* = 8)Daver et al. [29]Azacitidine + nivolumab + ipilimumabR/R AML24ORR: 46% CR/CRi: 36%/NADavids et al. [30]NivolumabAML and myeloid malignancies after transplant10 AML 19 myeloid malignancies/Median OS: 21.4 months 1-year OS: 56%DLTs 1 mg/kg (ir-AE): 33% (*n* = 2/6) DLTs 0.5 mg/kg (ir-AE): 18% (*n* = 4/22) ir-AEs: 9% (*n* = 2/22)fatal GVHD: 9% (*n* = 2/22)Ravandi et al. [31]Nivolumab + idarubicin + cytarabineND-AML and HR-MDS42 AML 2 HR-MDScomposite CR: 78%Median OS: 18.5 monthsEarly mortality: 5% irAEs grade 3/4: 13% 19 patients underwent allo-SCT GVHD: 68% (*n* = 13)Liu et al.[32]NivolumabMaintenance on AML in first CR, CR or CRi26/Median OS: 53.9 months; 2-year OS: 60.0%ORR: 32% Os rate 1 year: 56%Gojo et al.[34]Pembrolizumab + azacitidineNewly diagnosed AML and R/R AML37 R/R AML 22 de novo AMLORR: 55%, with 14% CR/CRi in R/R AML ORR: 94% with 47% CR/CRi in de novo AMLMedian OS for de novo AML: 13.1 monthsirAE n: 9 (24%) and 5 (11%) pts in Cohort 1, and 3 (14%) and 4 (18%) pts in Cohort 2Goswami et al. [35]Pembrolizumab + decitabineR/R AML10/Median OS: 10 monthsirAE n: 9. Trhombocytopenia: 80%. Neutropenia: 30%.Zeidner et al. [36]Pembrolizumab + high-dose cytarabineR/R AML37ORR: 48% with a composite CR 38%median OS: 13.2 monthsRare grade ≥ 3 ir-AEs after pembrolizumab: maculopapular rash (*n* = 2; 5%)Gao et al. [43] Tislelizumab + azacitidine or decitabine + CAG regimen (cytarabine, aclarubicin, G-CSF)R/R AML27ORR: 63%Median OS: 9.7 monthsGrade 2–3 ir-AEs: 14.8% (*n* = 4)Zeidan et al. [48]Arm A: Azacitidine + durvalumab 
Arm B: AzacitidineUntreated MDS or Elderly AML (≥65 year)129ORR Arm A: 31.3% Arm B: 35.4% 
DoR Arm A: 24.6 weeks Arm B: 51.7 weeksmOS: Arm A: 13.0 month Arm B: 14.4 monthArm A: constipation (57.8%) and thrombocytopenia (42.2%)Zeng [45]Avelumab + decitabineDe novo AML r/r AML7CR: 14% (*n* = 1) SD: 42% (*n* = 3)
febrile neutropenia (86%), hypoxia (57%), heart failure (29%), and pneumonitis (29%)Saxena [46]Avelumab + azacitidiner/r AML19CR/Cri: 10.5% median OS: 4.8 monthstrAEs grade ≥ 3: neutropenia, 10% (*n* = 2) Anemia: 10% (*n* = 2) irAEs grade 2/3: 10% (*n* = 2)

## 3. CTLA-4 Inhibition

CTLA-4, which is expressed on the surface of T cells, prevents T cell activation, and suppresses T cell maturation and differentiation. In T regs and naive resting T cells, CTLA-4 is expressed constitutively; it remains in the cytoplasm until activation, then it is upregulated on the surface within one or two days. In memory T cells, activation and expression occur even more quickly [49]. Preclinical studies in murine bone marrow chimeras hinted at the potential advantages of adjuvant CTLA-4 inhibition after the application of allo-SCT in AML. These studies demonstrated the robust anti-leukemic effects of these treatments, which also averted GVHD. Moreover, research performed on AML revealed the elevated expressions of co-stimulatory molecules (such as CD80 and CD86) that have been linked to a high risk of relapse and a trend towards poor prognosis [50,51].

### Ipilimumab

Ipilimumab is a human IgG1 kappa monoclonal antibody that specifically blocks CTLA-4. Davids et al. [51] postulated that ipilimumab can restore antitumor activity by promoting graft-versus-leukemia (GVL) mechanisms, and they rested this in a phase 1/1b study. In this trial, patients with different hematologic diseases that were in relapse after allo-SCT received ipilimumab at doses of either 3 or 10 mg/kg. The treatment was initially administered every 3 weeks for four cycles, and then again at 12-week intervals, for a maximum of one year. The study involved 28 patients, 12 of whom showed relapsed AML. Notably, ipilimumab at a dosage of 10 mg/kg yielded promising outcomes, with 5 out of the 12 AML patients achieving complete remission (CR). Among these responders, four showed extramedullary leukemia, and one had secondary AML, with three of the responses persisting for over a year. The responders showed a significant reduction in T regs and an increase in effector T cells’ activities, confirming the hypothesis of augmented GVL mechanisms [52]. The use of ipilimumab was assessed in combination with decitabine in a phase 1, multicenter, investigator-driven study (CTEP 10026). Patients with R/R MDS/AML were treated both with (Arm A) and without (Arm B) prior allo-SCT. The toxicity profile results were acceptable, with 50% of patients experiencing grade 1–2 ir-AEs, which were managed by steroid administration. Responses were observed in 8 of the 16 evaluable patients (3 CR, 2 CRi, and 3 marrow CR), with a median OS of 18.3 months [53]. In a high-risk MDS setting, the application of ipilimumab as a single agent showed poor activity, and did not improve outcomes. In a phase 1 study, among 29 HR-MDS patients who received ipilimumab, 1 patient reached the marrow CR with an ORR of 3.4%; the DOR here was only 3 months [54].

## 4. CD47-SIRPα Blockade

Malignant cells use the dominant macrophage immunological checkpoint CD47 to circumvent innate immunity. When CD47 binds to its receptor signal-regulatory protein alpha (SIRPα) on the surface of the macrophages, it transmits a “don’t-eat-me” signal, arresting phagocytosis. In 2001, the increased SIRPα in AML and its interaction with dendritic cells (DCs) and T cells [55] were reported for the first time. The CD47–SIRP interaction promotes the inhibition of myosin-IIA build-up at the level of the phagocytic synapse, as well as the recruiting of downstream Src homology-2 domain-containing protein tyrosine phosphatases (SHP-1 and SHP-2), thus preventing macrophage-mediated tumor phagocytosis [56]. Indeed, cellular stability requires a delicate balance between pro- and anti-phagocytic signals [57]. AML and HR-MDS have higher levels of CD47 than normal hematopoietic stem cells, contributing to the poor prognosis by enabling the evasion of phagocyte-mediated immune surveillance [58,59]. The Leukemia Stem Cell (LSC) is especially vulnerable to CD47 blockage, because it overexpresses calreticulin, a prominent pro-phagocytic signal [56]. Anti-CD47 monoclonal antibodies have been shown both in vivo and in vitro to reverse the anti- phagocytic signal and eliminate AML LSCs, suggesting their potential role in the treatment of AML and HR-MDS [60]. Several molecules that target the CD47-SIRP pathway are now being tested in clinical trials (Table 2) [61]. These novel agents act as either decoy receptors (SIRP-IgG Fc domain) or monoclonal antibodies that directly block CD47.

### Magrolimab and Other CD47/SIRPα Inhibitors

Magrolimab is a novel humanized immunoglobulin G4 anti-CD47 antibody that improves tumor cell phagocytosis by inhibiting the binding of CD47 to SIRPα [60]. In the phase 1 CAMELLIA trial, which involved 15 patients with R/R AML, magrolimab was well tolerated and exerted a modest single-agent activity. Of those enrolled, 73% achieved SD, and 40% showed lowered bone marrow blast counts (with a mean decrease of 27%) [62]. To enhance the anti-leukemic impact of CD47 inhibition, researchers have developed combination treatments involving drugs that stimulate the generation of signals promoting the engulfment of leukemic cells. This synergy can be used to trigger leukemic cell phagocytosis [63]. Preclinical investigations have demonstrated that azacitidine notably elevates the levels of calreticulin, a surface marker on malignant myeloid cells that stimulates their engulfment by phagocytes [63]. Magrolimab + azacitidine was also shown to boost phagocytosis in myeloid models both in vitro and in vivo, providing a solid mechanism justifying further explorations of the combination in patients [62,63,64].

A phase 1b trial (NCT03248479) was conducted to assess the efficacy and safety of magrolimab plus azacitidine in patients with AML who were unfit to receive intensive chemotherapy (*n* = 87). Here 82%, showed *TP53* mutations, among which 79% showed ELN-2017 poor-risk cytogenetics. Patients received magrolimab via a dose-escalation regimen (1 to 15 to 30 mg/kg) in combination with azacitidine. In total, 28 (32.2%) patients achieved CR, including 23 (31.9%) patients with *TP53* mutations. The median overall survival values in *TP53*-mutant and wild-type patients were 9.8 months and 18.9 months, respectively. The most frequent AEs were gastrointestinal, with half of patients developing nausea, constipation or diarrhea; 34.5% had anemia [65]. Based on these promising results, which are particularly interesting in the context of *TP53*-mutant AML (a category with very poor prognosis but treated only with specific drugs), there is an ongoing phase 3 ENHANCE-2 trial (magrolimab plus azacitidine vs. the physician’s choice of venetoclax plus azacitidine or 7 + 3 chemotherapy) focusing on untreated TP53-mutant AML (NCT04778397) [66], and trial ENHANCE-3 (NCT05079230) is focusing on patients with newly diagnosed AML who are ineligible for intensive hemotherapy.

Recently, Daver et al. presented data from a phase 1b/2 study (NCT04435691) that included a novel triplet regimen of magrolimab in combination with AZA and VEN in older/unfit or high-risk AML. The initial phase 1b trial enrolled only patients with R/R AML. The phase 2 expansion trial included both frontline and R/R patients. The frontline cohort enrolled patients ≥75 years of age, as well as patients with documented comorbidities leading to ineligibility for intensive therapy or harboring adverse-risk karyotypes (per the ELN 2017 criteria), and/or with *TP53* mutations regardless of age/fitness. The ORR value in de novo AML patients was comparable with values observed previously in patients receiving AZA/VEN, with an ORR of 80% (33/41), which includes an ORR of 74% (20/27) in patients with *TP53* mutations. Median OS was not reached in the ND, non-secondary AML patients; the median OS was 7.6 months among untreated secondary AML patients with a *TP53* mutation. Responses in the R/R AML group were scarce, with prior VEN-exposed patients faring poorly (ORR 11%), resulting in the discontinuation of this study arm. The grade 3 AEs included: anemia, which occurred in 23% (18/79), wherein the median hemoglobin drop was 1.2 g/dL after magrolimab infusion; febrile neutropenia (50%); pneumonia (38%); hyperbilirubinemia (11%); transaminitis (11%); creatinine elevation (8%) [67].

Evorpacept (ALX148) is a CD47-blocking protein with high affinity and a modified Fc domain that has been used in combination with magrolimab to prevent red cell agglutination. There are two ongoing trials assessing the use of ALX148 combined with AZA in high-risk MDS patients (ASPEN02 trial—NCT04417517) and with AZA + venetoclax in AML patients (ASPEN05 trial—NCT04755244). The combination of evorpacept + AZA seemed to be tolerable, according to the preliminary findings of the phase 1 portion of the ASPEN05 trial. No severe treatment-emergent AEs associated with evorpacept were seen—most that were seen were mild. Subjects with newly diagnosed and relapsed/refractory AML (including patients pre-exposed to venetoclax) showed an anti-leukemic activity, with bone marrow blast reduction ranging from 20 to 100% [68]. The second phase of the trial, comprising a random design, will further assess the combination.

A second-generation anti-CD47 IgG4 antibody, with a distinctive binding epitope exhibiting the benefit of reducing anemia, was created in China, and is known as lemzoparlimab (TJC4, TJ011133). A phase 2 clinical trial (NCT04202003) is being conducted on patients with R/R AML and MDS. One patient herein achieved a morphologically defined leukemia-free status, and only one out of the five enrolled patients experienced grade 3 AEs [69]. A phase 3 clinical trial combining lemzoparlimab and AZA has been authorized for HR-MDS patients (NCT05709093).
cancers-15-05060-t002_Table 2Table 2Trials including CD47/SIRPα inhibitors in patients with AML.ReferenceTherapeuticApproachNCT Number/ReferenceStageIndicationType of AMLNumber of PatientsAEsAvailable ResultsSurvivalNext Phase PlannedChao et al. [62]Magrolimab SANCT02678338 CAMELLIA trialPhase 1r/r AML HS MDS15 (NCT 20)Anemia 47% [70]SD: 73% Lower bone marrow blast counts: 40%

Daver et al. [65]Magrolimab + azacitidineNCT03248479Phase 1bAML unfit for intensive chemotherapy, naïve52Nausea, constipation or diarrhea, anemia (34.5%)*n* = 34 CR/Cri: 56% Of which MRD- (IF): 37% ORR: *TP53* Mutant AML: 71% (15/21) *TP53* mutant OS: 12.9 months *TP53* WT: 18.9 monthsYes phase 3 ENHANCE-2
Magrolimab + azacitidine or venetoclax + azacitidine or 7 + 3 (DA)NCT04778397 ENHANCE-2Phase 3AML untreated *TP53*-mutant87N/AN/AN/AOngoingDaver et al. [67].Magrolimab + venetoclax + azacitidineNCT04435691Phase 1/2Older/unfit or high-risk r/r AML74Grade 3 anemia in 23% (18/79)ORR de novo AML: 81% (35/43) *TP53* mutations: 74% (20/27) Responses in R/R AML were scarce (ORR 11%), this study arm closedMedian OS was not reached in newly diagnosed non-secondary AML patients; the median OS was 7.6 months among untreated secondary AML with *TP53* mutation
Ongoing [71]Evorpacept + azacitidineNCT04417517 ASPEN02 trialPhase 1/2High-risk MDS65 (planned)No severe treatment-emergent AEs Blast reductionNA
Garcia-Manero [68]Evorpacept + venetoclax + azacitidineNCT04755244 ASPEN05 trialPhase 1/2De novo AML r/r AML97 (planned)Data on 12neutropenia, anemia (6 each; 43%)Objective responses 6 ptsNAYesQi J. [69]Lemzoparlimab monotherapy or lemzoparlimab + azacitidineNCT04202003Phase 1/2R/R AML and MDS105 (planned) 8 with available dataNo SAE1 morphologic leukemia-free state



## 5. The TIM-3/Galectin9 Signaling

TIM-3, a co-inhibitory receptor, is present on various immune cells, such as CD4+ Th1 cells, CD8+ cytotoxic T-cells (CTLs), and in other innate immune cells like dendritic cells, monocytes, macrophages, mast cells, and NK cells. Additionally, it can also be found on cancerous cells [72,73]. The TIM-3 gene is located among the IL4 and IL-5 genes on chromosome 5q33.2 [74]. The most widely studied of the four TIM-3 ligands is galectin-9 (gal-9), which causes the apoptosis of Th1 cells [75] and is essential to tumor cell immune evasion. T cell dysfunction is caused by a TIM-3 overexpression in human and mouse tumor models [76]. TIM-3 and PD-1 are frequently co-expressed, and inhibiting TIM-3 by itself or in combination with other co-inhibitory molecules can reverse T cell exhaustion. TIM-3 cells have been intensively observed for their effects on immune cells, especially T cells and NK cells, which promote immunological exhaustion, as well as on LSCs, where they serve as a unique marker in the context of AML [77]. The ability of AML cells to proliferate was reduced when TIM-3/gal-9 binding was inhibited in vitro [78], while in mice, an anti-human TIM-3 MoAb eradicated LSCs without damaging regular hematopoiesis [79]. Several MoAbs, including MBG453 (sabatolimab), TSR-022, BMS-986258, LY3321367, SYM023, BGB-A425, and SHR 1702 [80], are under study for solid malignancies, but only MBG453 has demonstrated preliminary efficacy and safety in the context of AML and MDS so far. Eight phase 1/2 trials of MBG453 are now being conducted, either as a monotherapy or in combination with HMAs, PD-1 inhibitors, HDM201 (an MDM2 inhibitor), or venetoclax.

The preliminary data regarding the combination of MBG453 + HMAs (NCT03066648 trial) show ORR values of 58% in MDS and 38% in newly diagnosed AML patients in the MBG453 + DAC arm, and 70% in MDS and 27% in AML patients in the MBG453 + AZA arm, respectively. The most common AEs were thrombocytopenia, anemia and neutropenia; in the MBG453 + DAC group, four ir-AEs were reported (ALT increase, arthritis, hepatitis and hypothyroidism), while none were seen in the MBG453 + AZA cohort [81,82].

## 6. The LAG-3/MHC Pathway

The LAG-3 (CD223) gene is located on the short arm of chromosome 12 (12p13.32), and codes for a 70 kDa type I membrane CD4-like protein with higher affinity for binding to MHC class II than CD4, the engagement of which prevents T cell activation [1]. Given that the CD4 co-receptor is located nearby and has a similar intron/exon structure, it is likely that the CD4 and LAG-3 genes diverged from a single common ancestor gene as a result of gene duplication. The mature protein is composed of two metalloproteases (ADAM 10 and ADAM 17) that are activated by TCR signaling, a TM region with four external immunoglobulin-like domains, and a cytoplasmic tail that enables the intracellular transduction of inhibitory signals [83,84].

While information on AML is currently limited, the co-expression of LAG-3 and PD-1 in solid tumors has been associated with a poor response to PD-1 blockade, and has been employed as a biomarker for predicting the success of immunotherapy [85,86]. It should be kept in mind that immune suppression and antigen presentation mechanisms can both be impacted by MHC class II expression in AML blasts. Currently, researchers are exploring the use of PD-1 inhibitors, along with LAG-3-targeting antibodies, in the treatment of solid tumors, lymphomas, and multiple myeloma. The initial outcomes are promising, especially in the context of metastatic melanoma, where the combination of nivolumab and relatlimab has notably enhanced progression-free survival (PFS) [77].

The AARON study (NCT04913922) will test the safety and tolerability of relatlimab (anti-LAG-3) in combination with AZA and nivolumab in patients with relapsed/refractory AML and newly diagnosed AML, who are aged >65 years. This is the only study on the use of relatlimab in AML, and recruitment began in November 2022, with no results available at the time of writing.

## 7. The CD27/CD70 Axis

CD27 is part of the tumor necrosis factor (TNF) superfamily, and features a strong costimulatory molecule that is useful for T cell activation [87]. The CD27 signaling pathway is controlled by its distinct ligand, CD70, which is selectively increased on immune cells when they become activated. However, CD70 is not present in regular tissues or the hematopoietic system. This suggests that the initial stages of hematopoiesis do not rely on the CD27–CD70 interaction [88]. Abnormal CD70 expression alone (solid tumors) or together with CD27 co-expression (hematologic malignancies) favors the development of a suppressive microenvironment [89]. Indeed, CD27 expression enables immune escape, as high expressions of sCD27 are linked to a poor prognosis for AML, and abnormal CD27 expression can be found on LSCs of AML and chronic myeloid leukemia [90]. In LSCs, the CD27/CD70 pathway induces the aberrant activation of the Wnt pathway, leading to progression, drug resistance, and LSC proliferation [89]. Furthermore, the activation of the MEK pathway, the transcription factor AP-1, and the activation of beta-catenin (via the Wnt pathway) by CD27/70 can enhance the survival of AML cells. The anti-CD79 antibody cusatuzumab (previously ARGX-110), when tested in vitro, cleared LSCs by eliciting differentiation and apoptosis [91].

Based on these preclinical findings, a phase 1/2 clinical trial (NCT0030612) has been developed to assess the safety and efficacy of cusatuzumab, both as a standalone treatment and in combination with AZA, for AML patients who are not eligible for intensive therapy. A persistent response was obtained in six patients, in whom the median PFS was not reached, and the best hematological response was CR/CRi (two out of eight patients), with a median time to response of 3.9 months. Another study investigating cusatuzumab plus AZA in HR-MDS patients or newly diagnosed AML patients not eligible for intensive therapy has been completed, but the data are not yet available (NCT042415499). Moreover, there are currently two ongoing clinical studies, no longer enrolling participants, that are investigating the combination of cusatuzumab and AZA (the CULMINATE trial (NCT04023526)). This trial enrolled a total of 38 patients, with 12 in phase 1 and 26 in phase 2. Out of these patients, 19 achieved an objective response, defined as at least partial remission PR (50%), and 14 achieved a CR (36%) [92]. The study of using cusatuzumab + AZA + venetoclax (ELEVATE trial—NCT04150887) in newly diagnosed AML patients ineligible for intense chemotherapy is ongoing.

## 8. Bispecific Antibodies (T Cell Engager): BiTE

Bispecific T cell-redirecting antibodies are typically designed to bind to the CD3 chain, the signaling-invariant component of the TCR complex, and to a selected Tumor-Associated Antigen (TAA) [93]; in the context of AML, they are mainly directed against CD123 [94], CD33 [15], and CLL1 [95] (Table 3). The most relevant BiTE tested so far is flotetuzumab, an anti-CD123xCD3 that, in a phase 1 study on 88 patients with r/r AML after at least two lines of therapy, showed a 30% ORR. The AEs reported primarily comprised cytokine release syndromes (CRS) and infusion-related reactions, mostly of grade 1–2 [96]. Moreover, JNJ-63709178, a CD123xCD3-targeting antibody, was tested in a phase 1 study on 62 AML patients, with disappointing results characterized by minimal clinical activity (non-sustainable reduction in blasts) and substantial toxicity in terms of CRS, which is difficult to manage with step-up dosing [97]. In a phase 1 study of vibecotamab (XmAb14045), another CD123xCD3 BiTE, the preliminary data showed an ORR of 14% in AML patients [98]. Therefore, a phase 2 study (NCT05285813) on MRD-positive AML and MDS patients after HMA treatment failure is currenty recruiting at MD Anderson [99].

CD33 appears to be a feasible alternative target for BiTE treatment strategies. In a phase 1 study, AMG 330, an anti-CD33xCD3 with a short half-life (that requires continuous infusion, like blinatumomab), achieved evaluable responses in 8 of 42 (19%) patients: 3 CR, 4 CRi, and 1 morphologic leukemia-free state. The most frequent AEs included 67% showing CRS (grade ≥ 3 in 13%) and 20% showing nausea [100]. AMG 673 is similar to AMG 330, but is linked to an Fc molecule, which extends its half-life. In a phase 1 study of AMG 673, 12/27 (44%) evaluable patients showed a reduction in blasts compared with baseline, but only one patient achieved CRi, with an 85% reduction in bone marrow blasts [101]. AMV564 is a novel bivalent and bispecific (2:2) CD33xCD3 T cell engager (TCE), and was employed in a phase 1 study. Here, among 35 patients evaluable for efficacy, a reduction in bone marrow blast count was observed in 17 (49%), but only 1 CR and 1 CRi were attained [102]. These data discourage the further development of this treatment in hematologic malignancies, thus AMV564 is being explored (in combination with anti PD-1) for use in solid cancers [103].

In 2020, the last update was released regarding the ongoing phase 1 multinational study of the use of MCLA-117, a bispecific CLL1xCD3 TCE, in AML patients. Out of 26 evaluable patients with a post-baseline BM assessment, 4 (15%) showed a ≥50% blast reduction, including 1 with a morphological leukemia-free state [104]. These poor results are associated with CLL1 bimodality expression, which occurred in about 25% of the AML patients, hence limiting the effectiveness of CLL-1-targeted therapies [105].

Moreover, the use of bispecific Abs targeting FLT3xCD3 is under investigation in some early phase I studies; these include CLN-049 (NCT05143996) for use in the treatment of patients with R/R AML, and AMG 427 (NCT03541369).
cancers-15-05060-t003_Table 3Table 3Bispecific antibodies (T cell engager): BiTE in AML.NCT Number/ReferenceTargetTherapeuticApproachDevelopmental StageIndicationNo. of PatientsAEsAvailable ResultsSurvivalNCT02152956 VOYAGE Uy, G.L. et al. [96]CD123xCD3Flotetuzumab monotherapyPhase 1 Phase 1/2PIF/ER AML88 (246 planned)Cytokine release syndromes (CRS) and infusion-related reactions, mostly G 1–2CR/CRi 18.5% (5 of 27)mOS 10.2 monthsNCT02715011 Boyiadzis, M. [97]CD123xCD3JNJ-63709178 monotherapyPhase 1r/r AML62TEAEs ≥ 3 were observed in 11 (65%) patients. ISR: 100%. CRS: 27%.1 (1.6%) SD. Discontinued.NARavandi, F. et al. [98]CD123xCD3Vibecotamab monotherapyPhase 1LAM, (ALL-B, BPDCN)104CRS in 62 of 106 patients (85% grade 1–2, 15% grade ≥ 3)ORR del 14.8% 7/51. CR (2), CRi (3).
NCT05285813 Short, N.J. [99]CD123xCD3Vibecotamab monotherapyPhase 2AML MRD + MDS/CMML post-HMA13 (planned 40)N/A OngoingN/AN/ANCT02520427 Ravandi et al. [100]CD33xCD3AMG330 monotherapyPhase 1r/r AML MRD + AML MDS96CRS: 13% (67% ≥ grade 3) Nausea: 20%CR/CRiCRS: 9/42
NCT03224819 Subklewe et al. [101]CD33xCD3AMG 673 monotherapyPhase 1r/r AML30CRS: 50% AEs: 37% (50% grade ≥ 3) abnormal hepatic enzymes (*n* = 5, 17%), CRS (*n* = 4, 13%), leukopenia (*n* = 4, 13%), thrombocytopenia (*n* = 2, 7%), and febrile neutropenia (*n* = 2, 7%)Bone marrow blast reductions in 17 (44%),1 pz CRi
NCT3144245 [94,102],CD33xCD3AMV564 monotherapyPhase 1r/r AML36Grade ≥ 3 treatment-emergent AE: anemia, in 4 (11%)Bone marrow blast reduction in 17 patients (49%) 1 CR, 1 CRiN/ANCT03038230 [96], Mascarenhas J. et al.CD3xCLL-1MCLA-117 monotherapyPhase 1r/r AML newly diagnosed elderly untreated AMLHR-MDS50TEAEs were pyrexia (32%), CRS (32%), chills (22%), infusion site phlebitis (14%), vomiting (12%), and nausea (10%)Out of 26 evaluable patients, 4 showed ≥50% blast reduction including 1 with morphological leukemia-free stateN/A

## 9. Cellular Therapies (CAR-T)

Differently from cases of mature B cell non-Hodgkin lymphoma, acute lymphoblastic leukemia, and multiple myeloma, for which six second-generation chimeric antigen receptor (CAR) T-cell therapies have been approved by the FDA (idecabtagene vicleucel, lisocabtagene maraleucel, ciltacabtagene autoleucel, tisagenlecleucel, brexucabtagene autoleucel, and axicabtagene ciloleucel), none have yet been approved for AML.

The most striking difference is the lack of a specific target antigen, such as CD19, for AML, as this is exclusively expressed on the LSC and not in healthy hemopoiesis or other normal tissues. Many AML-specific antigens (e.g., CD33, CD123, CLL-1, CD70, and TIM-3) have been explored as targets for CAR T cells, as extensively reviewed elsewhere [106]. Consequently, the most common antigens targeted by CAR T cells so far include CD33 and CD123. A Phase 1 clinical trial (NCT03126864) sought to investigate the feasibility and safety of using autologous T cells, modified to express a CD33-targeted CAR T cells. Of the 10 patients assessed, only 4 carried CD33-CAR-T cells that met the pre-specified release criteria for infusion, and 1 died before re-infusion. Despite the effects observed (CRS ends the increase in IL6 and other inflammatory indexes), none of the patients met the criteria for response to treatment [107].

Also, CD123 has been used as a target in more than 20 trials of CAR T cells. In a study of seven patients with r/r AML, lymphocytes were electroporated and transfected with mRNA to transiently express a CAR T. All infusions of RNA CART123 were followed by a fever, with CRS following all but one infusion. Unfortunately, no blast reductions were seen [108].

In a phase 1 study of 12 pediatric patients (NCT04318678), CD123-CAR T cells were generated from CD4/CD8-selected autologous leukapheresis products using a lentiviral vector, which encoded a CD123-CAR. The two patients given DL1 showed no response. In the three patients on DL2, the following was observed: a reduction in blast percentage without complete remission (CR) in one patient, no response in one patient, and CR in one patient [109].

Therefore, CAR T cells are not broadly suitable for use in AML, as they feature several barriers yet to be overcome [18], and attempts have been made to produce off-the-shelf, allogeneic, natural killer (NK) cells, which represent an alternative cell population with unique properties that can also be modified to recognize specific proteins of AML (CAR-NK cells) [110]. Another interesting field of research concern T cell receptor (TCR)-redirected T cells, which can target the intracellular antigens presented by HLA molecules [111].

## 10. Conclusions

ITs have brought about a new era in solid cancer therapy, attaining unprecedented long-lasting responses even in metastatic diseases [1]. In the context of hematologic malignancies, similar results have been observed in lymphomas [2,3,4]. In the context of AML, although analogue studies have been conducted, most of them have failed to advance to the third phase, and no drug has yet been approved for AML.

Several explanations can be offered for the poor results attained in the context of AML. Indeed, cancer immune therapy, especially immune CIs (ICIs), have shown strong activity in tumors carrying a high mutational burden, such as melanoma, lymphoma, and myeloma, where the increase in the production of private neoantigens represents an ideal target for immune effectors. In contrast, AML, although presenting with molecular heterogeneity, falls within the category of low-mutational-burden tumors [105,106]. Moreover, AML bone marrow is a “tolerogenic” microenvironment, which can impair immune effectors’ activity even when they are boosted by ITs [112,113]. These observations may explain the failure of ICIs to “re-ignite” exhausted T cells in AML, although this T cell subset is abundant in the bone marrow of AML patients [109]. On the other hand, the presence of exhausted T cell populations represents a peculiar problem for TCE treatments, and alternative solutions, aside from combinations with ICIs and pauses in the administration of a TCE [114], are eagerly awaited.

Nevertheless, encouraging data on anti-CD47-blocking agents indicate that AML can be targeted by alternative ITs, which differentiates it from other tumors. Indeed, magrolimab has shown activity on *TP53*-mutated AMLs, an AML category with very poor prognosis. Anti-CD47-blocking agents are particularly effective when combined with innovative drugs in the context of AML therapy, such as HMA and BCL2 inhibitors; rendering them appealing for development in future clinical studies. Finally, the post-transplant setting needs to be further evaluated in ad hoc-designed trials, as this context could hypothetically magnify the power of ITs by promoting the “reset” of the tolerogenic bone marrow microenvironment and favoring boosted immune effectors, although at the putative cost of increased GVHD.

In conclusion, our knowledge on the efficacy of ITs in the context of AML requires further pre-clinical/clinical research, which will more clearly underline the patterns of immune escape/resistance shown by AML blasts. This will help in the design of new clinical studies, and maximize the current data, enabling us to successfully apply these innovative drugs in AML patients.

## Figures and Tables

**Figure 1 cancers-15-05060-f001:**
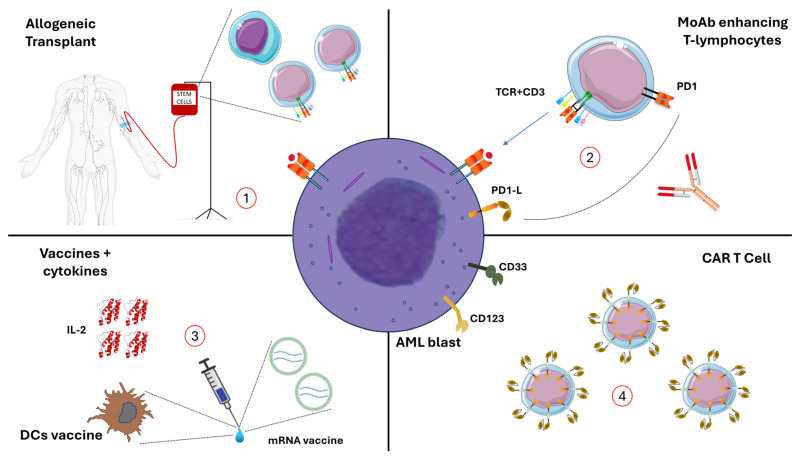
Different approaches to immunotherapy pursued in the fight against Acute Myeloid Leukemia. (**1**) The oldest and most established immunotherapeutic approach is represented by allogenic stem cell transplantation. Polyclonal T lymphocytes and HSC are injected into the patient. MHC presents leukemia-associated antigens to T lymphocytes, giving the first signal to the immunological synapse to enact the graft versus leukemia effect. (**2**) Exhausted lymphocytes might be reinvigorated using monoclonal antibodies, e.g., PD1-L and PD2-L can induce T cell anergy. Several types of anti-PD1 or anti-PD1L have been trialled in AML patients. (**3**) A previous system used to boost T lymphocytes is represented by the administration of cytokines like Interleukin-2 to enhance T cells. Another interesting strategy, currently under research, involves using vaccination to enhance AML antigens (mRNA vaccines are the most recently explored, but peptide vaccines and dendritic cell (DCs)-based vaccines have also been used). (**4**) CAR-T cells (autologous or allogeneic), transfecting T cells or NK cells have been actively investigated for use against AML antigens (CD33, CD123).

**Figure 2 cancers-15-05060-f002:**
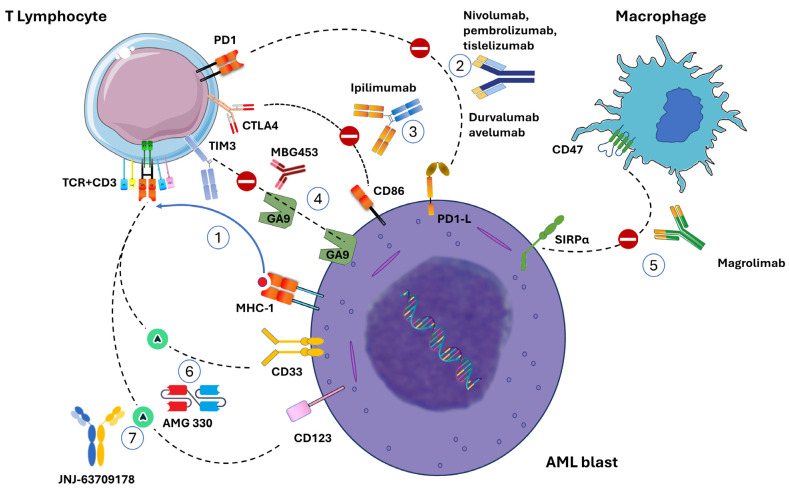
Immunotherapy with monoclonal antibodies in AML. (**1**) MHC presents leukemia-associated antigens to T lymphocytes, giving the first signal to the immunological synapse. (**2**) PD1-L and PD2-L can induce T cell anergy by linking the PD1 present on T lymphocytes. Several types of anti-PD1 (nivolumab, pembrolizumab, tislelizumab) or anti PD1L (durvalumab, avelumab) inhibit this crucial effect. (**3**) The CTLA4/CD86 axis can be inhibited by ipilimumab. (**4**) The TIM3/Galectin9 axis can be inhibited by sabatolimab and MBG453. (**5**) The SIRPα/CD47 axis presents the «don’t-eat-me» signal and can halt the phagocytosis of macrophages. Monoclonal Ab, similarly to magrolimab, can silence this signal and restore phagocytosis. (**6**) BITE AMG 330 is an anti-CD33xCD3. (**7**) JNJ-63709178 is a CD123/CD3-targeting antibody.

## Data Availability

Not applicable.

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
