# Peer review of "Immunotherapy with Monoclonal Antibodies for Acute Myeloid Leukemia: A Work in Progress"

_cancers, 2023, doi:10.3390/cancers15205060_

Round 1
Reviewer 1 Report
The paper "Immunotherapy of acute myeloid leukemia: a work in progressis" is well written and represents a significant contribution to the fild. I give credit to the authors for a comprehensive and very detailed review of all immunotherapy options in the treatment of AML.
The authors cite the majority of studies that have been conducted for each individual drug and concluding that most of them have failed to advance to phase 3 studies and currently no
drug is approved for AML, yet.
In order to make a large amount of datamore transparent, I suggest to make separate table, similar to Table 1, for each group of drugs. I would add to it the phase of the study and whether the next phases are planned or are currently being studied. It is also necessary to add unwanted events.
I advise revision of the figure. First of all, you need to add the name of the figure.
Consider adding the name of the drug group, drug to the figures themselves (next to the its mod of action). It would be extremely nice if there is a separate figure with the mechanism of action for each group of drugs.
Author Response
Separate tables were added for each class of drugs, with the revisions suggested.
Figure 2 was revised: the title of the figure was given; the Name of the drugs were added.
Reviewer 2 Report
This is a well written review addressing the recent developments in immunotherapy of AML. It would be important to mention and include the reports on the development of therapeutic vaccines and the T-cell therapies (CARs, TCR transgenic T cells).
The data on CD47 mAbs should be summerized in a Table as there are promissing results.
In addition, the authors should mention and include the publications by HJ Buhring as he was one of the first who described the binding of CD47 to SIRP-a and the expression on progenitor cells, AMLs and denritic cells (Seiffert M et al., Blood 2001, 97 (9):2741-9)
Author Response
The seminal work from Seiffert was cited and added. A table was added for anti CD47 drugs.
Reviewer 3 Report
The authors present an interesting review on the emerging role of immunotherapy in AML. However, some points of concern should be addressed:
11) It should be better clarified from the beginning what the authors mean with immunotherapy: indeed, the broader meaning would include allo-HCT, monoclonal antibodies such as rituximab, antibody-drug conjugates… thus enormous advances in the hematological field. I would start briefly explaining what immunotherapy is and its role in hematology. Then, I would clearly explain what the focus of the review is. The title might, therefore, be changed as well.
22) Other types of immunotherapy have been studied in the past (eg, IL-2) or are being developed in AML, although with several challenges (eg Vaccines, CAR-T cells) . While these might not be the main focus of the review, at least the latter group should be cited (Eg: Mardiana S, Gill S. CAR T Cells for Acute Myeloid Leukemia: State of the Art and Future Directions. Front Oncol. 2020; Coscia M, et al, Adoptive immunotherapy with CAR modified T cells in cancer: current landscape and future perspectives. Front Biosci (Landmark Ed). 2019). A table or a figure better describing the different immunotherapy approaches being explored could be useful.
English is globally clear and the paper well written
Author Response
The introduction was modified and included an explanation on the broader meaning of immunotherapy. The focus of the review was on monoclonal Ab. was better defined; the title was modified consequently. We added a small paragraph on CAR-T cells. We added a new figure 1 that describes the immunotherapy approaches.
Round 2
Reviewer 3 Report
the authors adequately adressed my concernes